# Lung Ultrasound After COVID-19: A Pivotal Moment for Clinical Integration—Navigating Challenges and Seizing Opportunities

**DOI:** 10.3390/healthcare13101148

**Published:** 2025-05-14

**Authors:** Damiano D’Ardes, Cristian Deana, Andrea Boccatonda, Daniele Guerino Biasucci, Francesco Cipollone, Mauro Castro-Sayat, Nicolás Colaianni-Alfonso, Adrián Gallardo, Luigi Vetrugno

**Affiliations:** 1Institute of “Clinica Medica”, Department of Medicine and Aging Science, “G. D’Annunzio” University of Chieti-Pescara, 66100 Chieti, Italy; damiano.dardes@unich.it (D.D.); francesco.cipollone@unich.it (F.C.); 2Department of Anesthesiology, Critical Care Medicine and Emergency, SS. Annunziata Hospital, 66100 Chieti, Italy; 3Department of Anesthesia and Intensive Care, Health Integrated Agency of Friuli Centrale, 33100 Udine, Italy; cristian.deana@asufc.sanita.fvg.it; 4Diagnostic and Therapeutic Interventional Ultrasound Unit, IRCCS Azienda Ospedaliero-Universitaria di Bologna, 40138 Bologna, Italy; andrea.boccatonda2@unibo.it; 5Department of Clinical Science and Translational Medicine, “Tor Vergata” University of Rome, 00133 Rome, Italy; biasucci@med.uniroma2.it; 6Intermediate Respiratory Care Unit, Juan A. Fernandez Hospital, Buenos Aires C1425, Argentina; nicolkf@gmail.com; 7Department of Kinesiology and Respiratory Care, “Sanatorio Clínica Modelo de Morón”, Morón C1015, Argentina; adriankgallardo@gmail.com; 8Department of Health Sciences, Kinesiology and Physiatry, National University of La Matanza, San Justo B1754, Argentina; 9Department of Emergency, Health Integrated Agency of Friuli Centrale, 33100 Udine, Italy; luigi.vetrugno@asufc.sanita.fvg.it; 10Department of Medical, Oral and Biotechnological Sciences, University of Chieti-Pescara, 66100 Chieti, Italy

**Keywords:** lung ultrasound, pneumonia, interstitial syndrome, ARDS

## Abstract

Lung ultrasound (LUS) has emerged as a valuable bedside decision-making tool, particularly since the COVID-19 pandemic, with applications in diagnosing pneumonia, managing fluid, and monitoring interstitial lung diseases (ILDs) and acute respiratory distress syndrome (ARDS), ultimately improving patient outcomes. Its repeatability, environmental safety, and reduced radiation exposure make it ideal for vulnerable populations and resource-limited settings. However, challenges such as inadequate documentation and a lack of standardized reporting formats limit its widespread adoption. The evolution of technology offers different possibilities, and improvements in software open up a range of possibilities, but this contrasts with the lack of postgraduate and undergraduate training and formal accreditation. This review addresses the impact of lung ultrasound through the course of air–liquid ratio impairment, crossing different clinical scenarios and exploring the challenges and opportunities for the implementation of lung ultrasound in the post-COVID era.

## 1. Introduction

Lung ultrasound (LUS) has significantly impacted clinical decision-making at the bedside over the past 30 years [1]. It is recognized as a disruptive innovation that has transformed healthcare delivery across various contexts, including acute care, pre-hospital settings, and follow-up services [2]. This imaging modality empowers healthcare practitioners to manage patients more effectively, endorsing a physiology-driven approach that has the potential to enhance patient outcomes [3,4]. Additionally, LUS is both repeatable and environmentally sustainable, drastically reducing patients’ exposure to ionizing radiation [5,6,7,8]. This feature is particularly beneficial in scenarios where alternative imaging techniques, such as computed tomography (CT) scans or chest X-rays, may not be readily available, such as in low- and middle-income countries, resource-constrained areas in high-income countries, and among vulnerable patient populations [9,10,11].

LUS has established itself as an ideal tool for the dynamic assessment of lung aeration in various pathologies and clinical scenarios. It proves valuable for both diagnosis and monitoring, correlating with diverse clinical outcomes [1]. However, the integration of LUS into clinical practice following the COVID-19 pandemic continues to be a matter of debate, further complicated by the existing gap in the literature regarding its application during this period.

This review discusses the influence of lung ultrasound throughout the progression of air–liquid ratio impairment, traversing multiple clinical scenarios and exploring the challenges and opportunities for the implementation of lung ultrasound in the post-COVID era.

## 2. Materials and Methods

A comprehensive literature search was performed by the primary author, encompassing the Medline, PubMed, ScieLo, and Google Scholar databases and utilizing “lung ultrasound”, “pneumonia”, “interstitial syndrome”, “interstitial lung disease”, and “ARDS” as pre-specified keywords and spanning the period from 1993 to 2024.

To ensure the relevance and methodological rigor of the included studies, three authors independently assessed the retrieved articles and selected those meeting the following criteria: (1) original research articles, systematic reviews, meta-analyses, and societal guidelines addressing the clinical use of lung ultrasound in adult patients diagnosed with ILD or ARDS and (2) studies assessing either the diagnostic accuracy of LUS, the correlation between LUS findings and clinical outcomes (e.g., mortality, length of stay), or the impact of LUS on clinical decision-making and patient management.

The following exclusion criteria were applied: (1) pre-clinical studies, including animal models; (2) studies primarily involving pediatric populations; (3) investigations focusing on imaging modalities other than LUS as the primary intervention; (4) articles not published in the English language; (5) editorials, letters to the editor, and conference abstracts lacking sufficient methodological detail or primary data.

## 3. Lung Ultrasound Basics

Ultrasound plays a crucial role in the diagnosis of interstitial lung diseases (ILDs), which encompass a diverse array of respiratory disorders that affect the interstitium and alveoli [12]. The secondary pulmonary lobule serves as the lung’s fundamental structural and functional unit, delineated by interlobular septa. These septa contain connective tissue fibers that extend toward the center of the lobule, forming an interstitial–alveolar space. Within this unit, bronchioles and arterioles are centrally located, while veins traverse the lobular walls, and the lymphatic system drains centrifugally toward the hilum and mediastinum [13,14].

In healthy lung tissue, the transmission of ultrasound (US) waves is limited due to significant disparities in acoustic impedance and sound wave propagation velocity between the tissues and the air. This results in reflection and reverberation artifacts [15]. However, these artifacts become more pronounced in pathological conditions, allowing for contextual interpretation [12,14,15]. Nevertheless, it is important to acknowledge its inherent limitations. One key limitation stems from the fact that LUS is primarily a surface-based imaging technique. Consequently, lesions that are not in direct contact with the pleura may be difficult to visualize or characterize adequately. Deep parenchymal lesions, for example, may be obscured by overlying aerated lung tissue, limiting the sensitivity of LUS for detecting these abnormalities.

Another significant limitation relates to the presence of factors that impede the transmission of ultrasound waves. Inadequate acoustic windows, such as those encountered in obese patients due to increased subcutaneous tissue, can significantly reduce image quality and limit the depth of penetration. Similarly, air interfaces, which strongly reflect ultrasound waves, can prevent or decrease the passage of ultrasound, hindering the visualization of underlying structures. This is particularly relevant in conditions such as pneumothorax, where the presence of air in the pleural space effectively blocks ultrasound transmission, making it difficult to assess the underlying lung parenchyma. Subcutaneous emphysema, characterized by air trapped within the subcutaneous tissues, can also create significant artifacts that obscure the underlying lung. These factors can limit the diagnostic utility of LUS in certain patient populations and clinical scenarios, necessitating the use of alternative imaging modalities [15,16].

## 4. Interstitial Syndrome and Interstitial Lung Diseases

US waves penetrate the pleura, generating artifacts associated with the loss of pulmonary aeration. Pathological increases in lung density may arise from various factors, including elevated extravascular lung water, collagen deposition, protein accumulation, blood infiltration, or the loss of lung aeration due to atelectasis [16].

Interstitial syndrome encompasses a range of disorders that involve the interstitial and/or alveolar regions of the lung, impairing alveolar–capillary gas exchange and potentially leading to varying degrees of respiratory failure. These conditions can present either acutely or chronically and may be categorized based on their distribution into focal or diffuse forms [17,18].

### 4.1. Interstitial Lung Diseases: Classification and Diagnostic Criteria

ILDs are characterized by the damage and thickening of the interstitium, which disrupts normal gas exchange. This damage often leads to symptoms such as progressive dyspnea and a persistent nonproductive cough [19]. ILDs can generally be classified into three main categories [20]:Idiopathic Interstitial Pneumonias (IIPs): Diseases of unknown etiology.Known-Cause ILDs: Specifically associated with identifiable factors or exposures.Secondary ILDs: Linked to other, less clearly defined processes or conditions.

### 4.2. The Crucial Role of LUS in Diagnosing Interstitial Lung Diseases

Lung ultrasound (LUS) has emerged as a valuable diagnostic tool for evaluating autoimmune interstitial lung diseases (ILDs). Its diagnostic efficacy and correlation with computed tomography (CT) images have been thoroughly documented [21]. A key feature identified through LUS is the presence of B-lines, defined as “laser-like vertical hyperechoic reverberation artifacts that originate from the pleural line, extend to the bottom of the screen without fading, and “move synchronously with lung sliding”. The identification of two B-lines per scan field is considered normal, whereas the presence of three or more lines is associated with varying degrees of interstitial syndrome.

The count of observed B-lines has been proven to correlate more effectively with disease severity than high-resolution CT scans, demonstrating its utility not only in diagnosis but also in severity stratification. The latest iteration of the LUS scoring system differentiates between moderate and severe loss of aeration based on the percentage of the pleura occupied by artifacts, which may include spaced or coalescent B-lines or subpleural thickenings. A loss of aeration is classified as moderate if 50% or less of the visualized pleura is affected; conversely, if the affected area is clearly greater than 50%, it is classified as severe [22]. See Figure 1 and Appendix A.

Additionally, increased irregularity of the pleural line and greater distance between B-lines have been associated with more severe fibrosis and worsened pulmonary physiological parameters, such as total lung capacity (TLC) and diffusing capacity of the lung for carbon monoxide (DLCO) [23,24,25]. In fact, research has also shown that the quantity of B-lines is inversely correlated with pulmonary function test (PFT) results and DLCO, indicating that B-lines possess both diagnostic value and pathophysiological significance, which may be crucial for ongoing monitoring [23,24,25,26].

The main ultrasound signs for identifying ILD include the following [27,28]:Multiple B-Lines: These appear in a diffuse, inhomogeneous distribution and are critical for ILD diagnosis.Pleural Line Abnormalities: These may include thickening, irregularities, and fragmentation of the pleural line.Subpleural Abnormalities: Small echo-poor areas observed beneath the pleural line also indicate ILD. See Figure 2 and Appendix A.

### 4.3. Impact of LUS on SARS-CoV-2

The SARS-CoV-2 pandemic has highlighted the significance of B-lines in diagnosing interstitial viral pneumonia [29]. The presence of diffuse, multifocal, and bilateral B-lines indicates interstitial–alveolar pneumonia, with severely affected areas presenting as consolidations. Additionally, pleural effusion may be observed in some cases [30,31,32]. COVID-19 pneumonia exhibits distinct ultrasound characteristics, including an irregular and thickened pleural line, as well as small, subcentimeter subpleural consolidations that appear hypoechoic [32,33,34]. LUS has proven to be an effective and reassuring follow-up tool, demonstrating a reduction in B-lines and the reappearance of A-pattern lung fields following pharmacological treatments [35,36,37]. Furthermore, LUS has demonstrated its value in monitoring both IMV and NIRS, including in the awake prone position, showing associations with various relevant clinical outcomes [38,39,40,41,42,43].

In summary, interstitial syndrome, characterized by the thickening of the interlobular septa and the presence of B-lines, alongside other lung diseases, represents a complex interplay of pathological processes that can be effectively diagnosed and monitored using lung ultrasound. The identification of B-lines, in conjunction with other ultrasound findings, is essential for the diagnosis, assessment, and management of these conditions.

## 5. LUS for the Detection of Consolidations

Lung consolidation occurs when the lung parenchyma loses air, resulting in a profound de-aeration process. Diseases that lead to fluid accumulation—be it transudate or exudate—typically begin by affecting the interstitial level before progressively impacting the alveolar space. This progression results in a significant reduction in alveolar air content. When the pulmonary cortex attains a specific density, classified as below 10% of normal air content, the ultrasound beam can penetrate the tissue, generating a “real”, non-artifactual grayscale image of varying extension [44].

### 5.1. Limitations

On ultrasound, consolidated lung tissue presents as a hypoechoic subpleural structure, exhibiting a tissue-like appearance. The pleural line is typically interrupted or demonstrates reduced echogenicity [45]. Furthermore, posterior acoustic enhancement is often observed, which is characteristic of bronchopneumonia [44].

It is essential to acknowledge that ultrasound may occasionally underestimate the size of lesions, particularly when they are associated with sub-consolidation thickening, which can lead to a satellite interstitial syndrome. This syndrome, a component of the inflammatory process, presents an artifactual nature that complicates the accurate definition of the true extent of the condition. In instances of extensive consolidation, an entire lung lobe may appear consolidated, closely resembling a hepatic segment, a phenomenon referred to as “hepatization” [46]. See Figure 3, Figure 4, Figure 5 and Figure 6 and Appendix A.

### 5.2. Differential Diagnosis

The differential diagnosis for pulmonary consolidations is extensive, encompassing conditions ranging from pneumonia to tumors. The primary distinguishing factor lies in the distribution of the lesions: multifocal and bilateral lesions are typically associated with infectious conditions or acute respiratory distress syndrome (ARDS), whereas unilateral lesions are more indicative of neoplastic processes. Numerous studies have aimed to identify semiological signs for differential diagnosis, particularly through the assessment of internal bronchograms [47,48].

Air bronchograms manifest as reverberant echogenic streaks reflecting residual air within the bronchial tree in the area of consolidation. They can be classified into static and dynamic categories. Static air bronchograms may be observed in cases of compression atelectasis or incomplete atelectasis, while dynamic air bronchograms, which move with respiration, are suggestive (but not definitive) of infectious pneumonia [46,47,48,49]. Fluid bronchograms consist of tubular structures filled with hypoechoic content, clearly displaying their walls without a color Doppler signal, thus aiding in their differentiation from vascular structures [45,46,50,51]. These bronchograms indicate a complete loss of air within the consolidation and are often associated with inflammatory processes characterized by excessive mucus production in children. In adults, they may be linked to obstructive atelectasis due to tumor masses [50,51].

Infectious consolidations typically present a non-homogeneous hypoechoic appearance, with dynamic air bronchograms observed in 80–90% of cases [52]. The pleural line is generally interrupted or altered, with corresponding reduced or absent sliding. Parapneumonic pleural effusion is often present, and color Doppler investigations may reveal triphasic flows with a resistance index greater than 0.8, indicative of pulmonary arterial vasculature.

Pulmonary infarctions usually appear as homogeneous, hypoechoic, wedge-shaped or polygonal/triangular consolidations, primarily located in the dorsal regions of the lower lobes, and often seen in multiple instances [53,54]. On color Doppler examination, internal vessels are typically not visible, making it rare to identify a congested embolized vessel (referred to as the vessel sign) as a transonic tubular structure at the apex of the consolidation. Additionally, pleural effusions may be present adjacent to the consolidation [54].

Neoplastic lesions can be characterized as non-homogeneous, oval, or polycyclic hypoechoic subpleural consolidations, accompanied by an interrupted pleural line, posterior echogenic reinforcement, and often anechoic necrotic areas located centrally. Signs of local infiltration into the pleura and chest wall may also be evident. Dynamic air bronchograms are less commonly seen in neoplastic lesions. The pleural line is typically interrupted, and posterior echogenic reinforcement may be observed. A key differentiating feature is the presence of anechoic necrotic areas located centrally within the lesion, suggesting tissue death. Signs of local infiltration into the pleura and chest wall, such as pleural thickening or direct extension of the lesion, may also be evident, indicating aggressive behavior. While pleural effusions can occur with both, neoplastic effusions are more likely to be complex or loculated [55,56].

### 5.3. Promising Tools

Recent studies have investigated innovative ultrasound techniques to differentiate among various lesions. Contrast-enhanced ultrasound (CEUS) shows promise, although it has not yet been widely adopted in the field of pneumology. Its primary applications include evaluating suspected pulmonary infarctions, distinguishing between neoplastic masses and fibrinous or cicatricial areas within the pleural cavity, and facilitating biopsy procedures [56,57].

Additional research has explored elastosonography as a method to differentiate malignant from benign lesions, based on the premise that tumors exhibit greater tissue stiffness [57,58,59]. Moreover, artificial intelligence technology holds the potential to enhance diagnostic accuracy across a range of pathologies [60,61].

## 6. Acute Respiratory Distress Syndrome (ARDS)

ARDS is characterized by non-cardiogenic pulmonary edema, leading to impaired gas exchange and short-circuiting in the alveoli [62,63]. Recent studies show a strong correlation between LUS findings and the typical progression of ARDS. While LUS is still being researched and has not universally replaced traditional diagnostic techniques, it shows considerable promise as a first-line diagnostic tool, especially in intensive care units (ICUs). During the first week after pulmonary or extrapulmonary injury, the interstitial and alveolar spaces fill with protein-rich fluid [64]. This exudative phase transitions to the proliferative phase starting in the second week, and from the third week onward, the lungs may undergo fibrosis or resolution of ARDS [65].

As lung aeration diminishes, lung ultrasound (LUS) patterns transition from A to B. B-lines appear as discrete, vertical, laser-like hyperechoic lines that originate from the pleural surface, extending to the bottom of the screen without fading [64]. Coalescing B-lines signify substantial pulmonary aeration loss due to partial filling of the alveolar spaces, while pulmonary consolidation arises from a more extensive loss of aeration. This is visualized as a tissue echotexture with a distinct boundary at the pleural line and an irregular boundary with the aerated lung, often referred to as the “shred sign” [65]. Additionally, the presence of air bronchograms may be observed [47,48,49].

The initial proposal for utilizing LUS to diagnose ARDS is anchored in the Kigali modification of the Berlin definition [66]. This modification was specifically designed for resource-limited settings where access to CT scans and mechanical ventilation is constrained. According to the Kigali modification criteria, the identification of bilateral B-lines or consolidations via LUS can satisfy the imaging criteria for ARDS. In a random sample of 54 out of 1069 ultrasounds (5.1%) performed independently by two operators, the interobserver agreement was 96.3% (classifying bilateral B-lines or consolidation with effusion versus unilateral findings or none), yielding a kappa statistic of 0.92 (95% confidence interval, 0.82–1.00). LUS in Kigali’s modification is based on findings (bilateral B-lines or consolidations) to satisfy the imaging criteria for ARDS, offering a simplified approach to diagnosis when advanced imaging is unavailable. This score is primarily qualitative, focusing on the presence or absence of specific LUS findings for diagnostic purposes. However, when compared to CT scans in high-income countries, these criteria demonstrated high sensitivity but low to moderate specificity. Nevertheless, they were integrated into the global definition of ARDS [67].

### 6.1. Sonographic Findings

Multiple studies have indicated that lung ultrasound (LUS) findings closely correlate with those obtained from computed tomography (CT), which is considered the gold standard for the assessment of acute respiratory distress syndrome (ARDS). For instance, diffuse B-lines, pleural irregularities, and consolidations identified via LUS often correspond to ground-glass opacities and dependent consolidations observed in CT scans. Despite CT being the gold standard, LUS demonstrates high sensitivity and specificity for detecting interstitial–alveolar syndrome, with imaging abnormalities in ARDS characterized by a heterogeneous presentation.

Sonographic signs indicative of ARDS include the following [68,69]:Anterior subpleural consolidations;Absent or reduced lung sliding;“Preserved areas” of normal lung parenchyma;Abnormalities in the pleural line (e.g., fragmented, thickened, or irregular pleural line);Inhomogeneous distribution of B-lines.

One limitation of LUS is that it primarily serves as a superficial imaging technique, which may not effectively visualize lesions located deeper within the lung if a normally aerated parenchyma exists between the probe and the lesion [70,71].

### 6.2. LUS as a Tool for Intervention Assessment and Outcomes

LUS has been effectively utilized in the management of ARDS for both positive end-expiratory pressure (PEEP) titration [65,66,68] and monitoring lung aeration and its association with the failure of NIRS [42,72]. It can also aid in predicting ARDS risk in certain scenarios by quantifying lung contusion and identifying patients likely to develop ARDS [73]. One notable advantage of LUS is its ability to be performed repeatedly without risk to the patient, enabling clinicians to monitor the progression or resolution of ARDS in real time. This feature is particularly beneficial for adjusting ventilator settings or assessing patient responses to prone positioning [39,73]. Moreover, during the COVID-19 era (COVID-ARDS), lung ultrasonography illustrated its role in severity stratification, alongside associations with mortality and the need for intensive care [42,72,73,74,75].

### 6.3. The LUS-ARDS Score System

A recent multicenter prospective observational study introduced a novel scoring system known as the LUS-ARDS score. This score is based on lung ultrasound (LUS) aeration scores for both the left and right lungs, along with anterolateral pleural line abnormalities. The performance of the LUS-ARDS score was compared to chest X-rays in patients with ARDS, demonstrating an area under the receiver operating characteristic (ROC) curve of 0.90 (CI 0.85–0.95). This performance is comparable to that of experienced chest X-ray readers, with the LUS-ARDS score exhibiting superior diagnostic accuracy at each cut-off point [76]. In this scoring system, identifying pleural abnormalities is essential for distinguishing ARDS from cardiogenic edema. Moreover, the integration of LUS with cardiac ultrasound should be considered [77]. This score represents a more quantitative approach that assesses lung aeration scores for both the left and right lungs, along with anterolateral pleural line abnormalities. This score assigns numerical values to different LUS patterns (e.g., normal aeration, B-lines, consolidation) and calculates a total score that reflects the overall severity of lung injury. The LUS-ARDS score has been shown to correlate with clinical outcomes and may be useful for monitoring disease progression and response to therapy [76].

### 6.4. The LUS Score as a First-Line Screening Tool

LUS has proven effective in assessing lung aeration and its relationship with respiratory system compliance (Crs) [78] and mechanical power (MP) [79], as well as identifying ARDS phenotypes [80]. This is of particular interest, as failure to correctly identify phenotypes and attend to lung morphology may increase mortality [81]. Notably, ultrasound-measured lung aeration in both adults and neonates, regardless of the presence of acute respiratory failure, appears comparable. Furthermore, lung aeration assessed using the LUS score is significantly and inversely correlated with respiratory system compliance, irrespective of the patient’s age, indicating that a higher LUS score correlates with a reduced volume available for tidal ventilation [78]. In the context of ARDS and respiratory failure, LUS plays a vital role in monitoring patients and their ventilation, enabling physicians to tailor mechanical ventilation strategies based on LUS findings [79,80,81,82,83,84].

Recent research suggests that while low oxygenation correlates with poor lung aeration, mechanical power (MP) and driving pressure (DPaw) are more closely linked to ventilation settings and the size of the “baby lung” (the well-ventilated lung areas). In fact, higher MP and DPaw values correspond to increased ventilation intensity and poorer lung mechanics, regardless of oxygenation levels [85]. In fact, PaO2/FiO2 shows inconsistent correlations with clinical outcomes [85] and may not even correlate with mortality in early classic ARDS [86]. Reduced physiological dead space, rather than PaO2/FiO2 (insensitive to alveolar overdistension), correlates more strongly with survival and improved respiratory mechanics [87]. This is relevant because lung ultrasound (LUS) effectively detects lung recruitment [88], and this is associated with reduced dead space; however, direct comparisons between LUS-measured aeration changes, respiratory mechanics, and dead space are lacking. Likewise, ventilator maneuvers to identify recruitable from nonrecruitable lungs have not been formally compared with regional changes in aeration by LUS [89].

Based on current evidence and clinical practice, LUS may serve as a first-line screening tool for ARDS in the ICU, particularly in unstable or resource-limited settings. However, its role as a definitive diagnostic instrument has yet to be firmly established, and there remains a lack of universal diagnostic criteria for ARDS based on LUS findings. While typical presentations, such as bilateral B-lines, are well documented, no standardized protocol or diagnostic algorithm exists for diagnosing ARDS solely through LUS.

The accurate diagnosis of ARDS often requires the exclusion of cardiac causes of pulmonary edema, a task that may not be fully accomplished with LUS alone, particularly in the absence of concurrent echocardiography. Until standardized protocols are widely adopted and validated, LUS should be integrated into a multimodal diagnostic framework that includes clinical assessment, echocardiography, and confirmatory imaging with CT scans when feasible.

The future of LUS appears promising, especially with the potential incorporation of machine learning algorithms to enhance diagnostic accuracy and reduce operator variability. Such advancements could position LUS as a cornerstone in the diagnosis of ARDS in the near future. See Table 1.

## 7. Controversy and Future Perspectives

### 7.1. Difficulties in Implementation in Daily Practice

First, it is crucial to differentiate between two types of LUS examinations: emergency LUS, which is performed to exclude acute conditions, and elective LUS, intended to be as comprehensive as possible. The primary objective of the emergency examination is to deliver a swift assessment (Yes/No) without unnecessary details. In contrast, an elective and thorough LUS examination should encompass critical elements such as the date, day, and time of the exam; the techniques employed and the type of probe used; the protocol followed for the study; the number of scanning sites; and the patient’s clinical condition at the time of the exam. In this context, a meticulous and detailed account of the ultrasound findings is essential [90].

One significant issue is that physicians rarely document their examination results in medical records, partly due to the absence of a validated format for reporting findings. This underutilization highlights the urgent need for change and innovation in integrating LUS into clinical practice. The absence of a standardized reporting format significantly diminishes the practical utility of LUS in routine clinical settings. Unlike chest X-rays and CT scans, LUS results are often neglected in the patient’s clinical records, if they are recorded at all [90]. The aphorism “translation is betrayal” aptly illustrates the difficulty of fully conveying concepts through language.

Second, training in LUS is a subject of considerable debate. As a relatively new and technical field, it currently lacks validated certifications specifically tailored for the pulmonary ultrasound curriculum. Additionally, the classification of skill levels—such as basic, intermediate, and advanced—remains poorly defined, especially compared to established ultrasound techniques like echocardiography. This ambiguity has led to significant inconsistencies in training pathways within this critical area [91].

Many experts advocate for a systematic approach to acquiring US skills, emphasizing the importance of incorporating training into both undergraduate and postgraduate medical education. The absence of formal certification, particularly for point-of-care ultrasound (POCUS) in critical environments, is seen by both specialists and trainees as a substantial barrier to skill development [92,93].

### 7.2. Future Perspectives

Training in lung ultrasound has been effectively integrated into broader point-of-care ultrasound certification programs in several countries, starting from undergraduate education and extending through residency and fellowship training. This alignment facilitates the mutual recognition of competencies across different medical contexts, enhancing educational outcomes and fostering a cohesive framework for educational and certification programs in LUS [92,93].

The incorporation of artificial intelligence (AI) technologies in the training and application of lung ultrasound holds significant potential to transform the field by improving diagnostic accuracy, reducing operator variability, and optimizing training processes. By leveraging machine learning algorithms, educational programs can be tailored to individual learning paces, providing targeted feedback to trainees. Additionally, AI systems can assist in interpreting ultrasound findings, enabling healthcare providers to make more informed clinical decisions. Recent studies have demonstrated that the application of AI to lung ultrasound has significantly increased the efficiency of non-expert ICU operators, underscoring AI’s potential to enhance effectiveness and reduce interpretation times [60,61].

Research indicates that AI algorithms, particularly deep learning techniques, can improve image acquisition and interpretation across various ultrasound applications, including pulmonary diagnostics, streamlining diagnostic workflows and making ultrasound more efficient. This technology can also play a crucial role in resource-limited settings by enabling rapid diagnoses in point-of-care environments while alleviating the burden on healthcare professionals by reducing the time spent on image analysis, thus allowing for a more efficient allocation of medical resources. Furthermore, the promise of AI in ultrasound extends to the development of educational programs that leverage these technologies, preparing future practitioners to effectively utilize advancements in imaging technology [94].

However, it is essential to approach the increasing prevalence of these advanced technologies with caution. The ongoing debate regarding the appropriate role of AI in clinical practice necessitates defining how and when AI should be integrated into diagnostic processes and education. While AI undoubtedly plays a significant role in contemporary medicine, establishing its roles and limitations is vital to ensure responsible and effective utilization in the future. Consequently, educational institutions must incorporate AI into their curricula to adequately prepare future practitioners to leverage these advancements effectively.

### 7.3. Strengths and Limitations

This review benefits from a comprehensive literature search across multiple databases (Medline, PubMed, ScieLo, Google Scholar) and a clearly defined search strategy using pre-specified keywords. The independent assessment of retrieved articles by three authors enhances the rigor and reduces potential bias in study selection. Furthermore, the review addresses a timely and clinically relevant topic, exploring the integration of LUS in the post-COVID-19 era and the potential impact in the near future.

However, several limitations should be acknowledged. First, the restriction to articles published in English may introduce selection bias, potentially excluding relevant studies and perspectives from non-English speaking regions. Second, the evolving nature of both LUS techniques and AI applications in several lung diseases and scenarios means that the evidence base is still developing, and future research may alter the conclusions presented here.

Despite these limitations, this review provides a valuable overview of the current state of LUS, highlighting both the opportunities and challenges associated with their integration into clinical practice. Further rigorous research is needed to establish standardized protocols and diagnostic criteria, particularly in the application of AI technologies, and to address the practical considerations for widespread implementation.

## 8. Conclusions

Lung ultrasound (LUS) has emerged as a critical tool in diagnostics for lung diseases like pneumonia, ILDs, and ARDS, particularly for the assessment of air–liquid ratio impairment. The increasing body of literature supporting its utilization underscores its clinical efficacy and relevance. However, challenges such as heterogeneity in training protocols and a lack of standardized applications must be addressed to ensure effective implementation in acute care settings. By establishing formal education and certification programs, we can enhance the competency of healthcare providers in LUS execution within clinical practice. Furthermore, the integration of artificial intelligence into LUS applications has the potential to optimize diagnostic workflows and augment accuracy, thereby revolutionizing pulmonary care delivery in the post-COVID era. The confluence of rigorous training and advanced technology will be essential in fortifying the role of LUS as a cornerstone in the management of respiratory conditions in intensive care settings.

## 9. Key Messages


Lung ultrasound (LUS) has emerged as a vital tool in assessing post-COVID-19 lung diseases, enabling rapid, safe, and accessible diagnosis, particularly in resource-limited settings.LUS provides notable advantages over other imaging methods, including its repeatability, absence of ionizing radiation, and suitability for vulnerable populations such as children and pregnant women.To enhance its integration into everyday clinical practice and optimize medical decision-making, the standardization of protocols and systematic documentation of ultrasound findings are essential.Lung ultrasound is crucial for diagnosing and monitoring severe conditions like ARDS, ILDs, and pneumonia, as it can effectively identify B-lines, consolidations, and pleural alterations.The future of LUS is being shaped by advancements in artificial intelligence and standardized training, which will improve diagnostic accuracy, reduce variability between operators, and broaden its use across various clinical settings.


## Figures and Tables

**Figure 1 healthcare-13-01148-f001:**
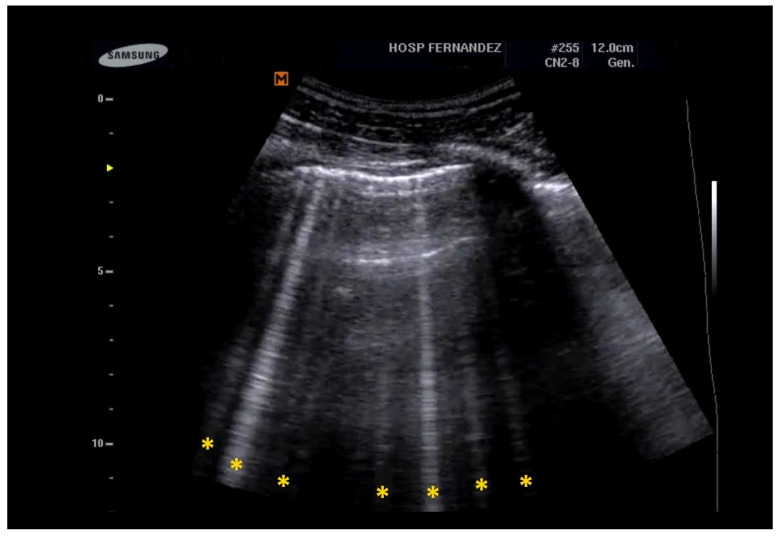
B-lines are hyperechoic artifacts descending from the pleural line to the bottom of the screen (yellow *).

**Figure 2 healthcare-13-01148-f002:**
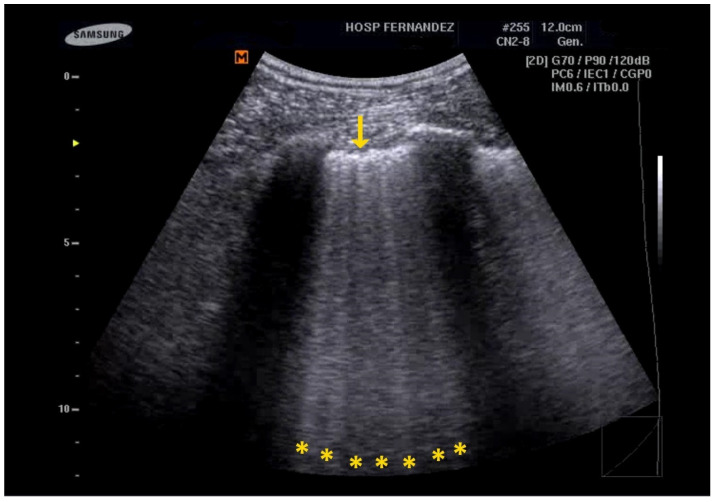
Multiple B-lines and an irregular pleural line (yellow * and yellow ↓).

**Figure 3 healthcare-13-01148-f003:**
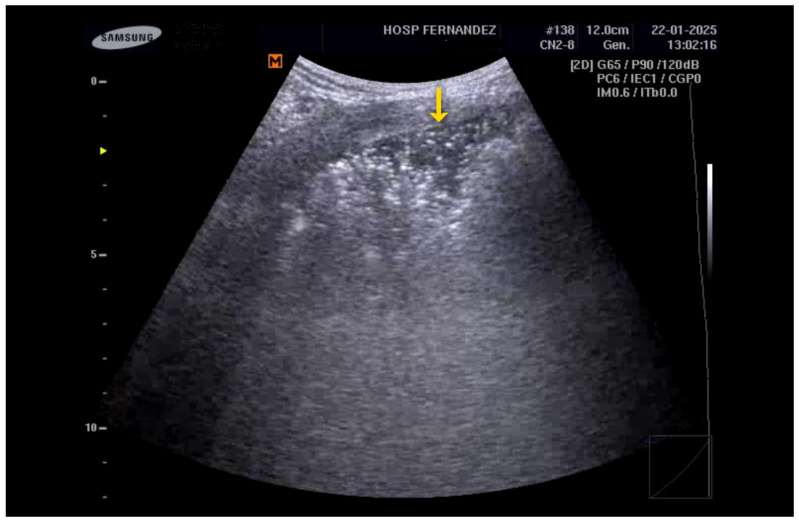
Subpleural consolidation. “Shred sign” (yellow ↓).

**Figure 4 healthcare-13-01148-f004:**
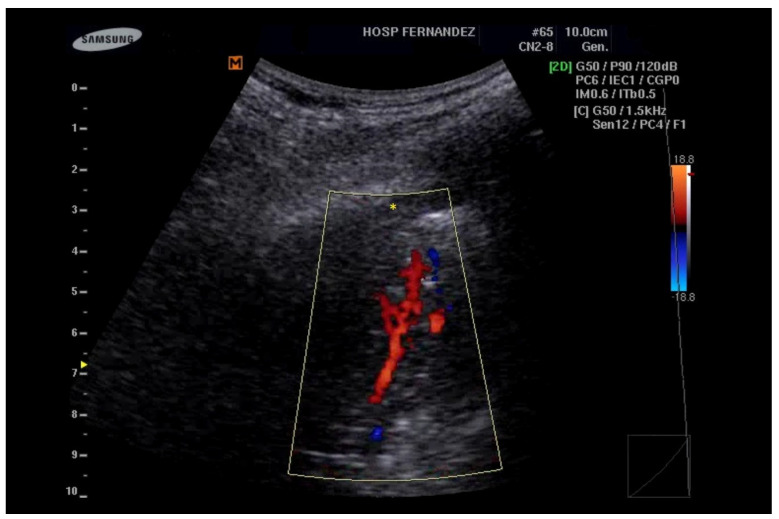
Consolidations with an LUS score of 3 indicate a complete loss of aeration measuring greater than 2 cm (yellow *). In addition, the preservation of the vasculature can be detected by color Doppler.

**Figure 5 healthcare-13-01148-f005:**
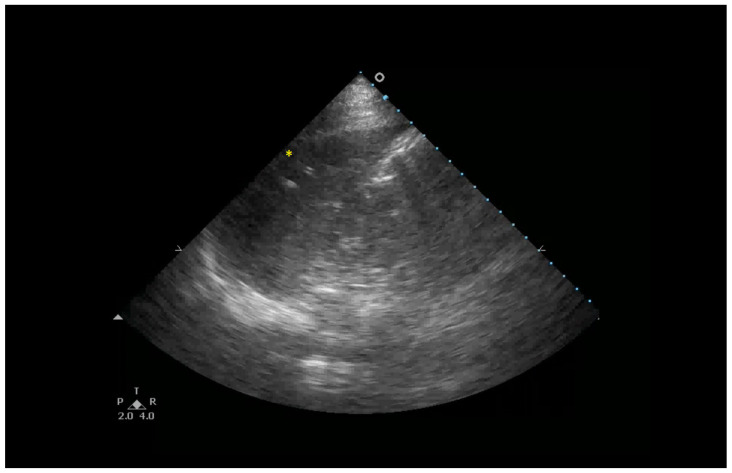
Consolidations with an LUS score of 3 indicate a complete loss of aeration in the entire inferior right lobe (yellow *).

**Figure 6 healthcare-13-01148-f006:**
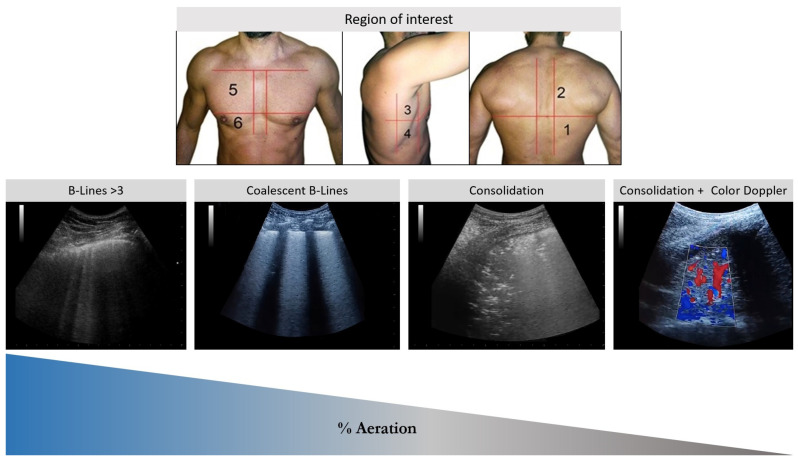
Ultrasonographic patterns of lung aeration and region of interest. Fewer than three well-spaced B-lines or coalescent B-lines on <50% of the visualized pleura indicating a mild loss of aeration. Coalescing B-lines: coalescent B-lines with or without subpleural consolidations on >50%, indicating a moderate loss of aeration. Consolidation: a transition between coalescent B-lines and a weaving or consolidation pattern, which indicates moderate to severe loss aeration. Consolidation + color Doppler: a large consolidation > 2.5 cm in diameter, indicating a severe loss of aeration with preserved blood flow.

**Table 1 healthcare-13-01148-t001:** Summary of the most relevant findings in different clinical scenarios in lung ultrasound.

Clinical Scenarios	B-Lines Distribution		Pleural Line		Consolidation			
	Homogeneous	In-homogeneous	Irregular or fragmented	Thickening	Small echo poor aeration areas	Consolidation	Air bronchogram	Visible lung vasculature on Doppler
ILDs	NO	YES	YES	YES	YES	NO	NO	NO
Pneumonia	NO	YES, if present	YES	NO	NO	YES	YES	YES
ARDS	NO	YES	YES	YES	YES	YES	YES (conditional to severity)	YES (conditional to the severity)

ILDs: interstitial lung disease; ARDS: acute respiratory distress syndrome.

## Data Availability

The data can be requested from the corresponding author.

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
