# Peer review of "Lung Ultrasound After COVID-19: A Pivotal Moment for Clinical Integration—Navigating Challenges and Seizing Opportunities"

_healthcare, 2025, doi:10.3390/healthcare13101148_

Round 1

Reviewer 1 Report

Comments and Suggestions for Authors

I appreciate the opportunity to review this paper. It addresses a topic of great relevance in the field of ultrasound as difficulties in its implementation in daily practice. It is a nice review of the principles and basic lesions you can find when performing LUS in different pathologies. It is a profound and updated review of the literature available about LUS. However, I feel that there are some essential aspects that could be improved to increase the impact and clarity of the manuscript.

Title: The current title does not adequately reflect the content nor does it provide an accurate idea of what the reader can expect to find in the text. I suggest rephrasing the title so that it better captures the focus and main objectives of the paper.

Objectives: The objectives are not sufficiently defined and, moreover, do not strictly correspond to the development of the manuscript. In the abstract,  the author presents this as objective:  “to advocate for the ongoing integration of LUS in routine clinical practice, particularly for diagnosing and monitoring lung diseases, and to emphasize its significant impact on patient outcomes following the COVID-19 pandemic”. Later on in methods it stated that “This narrative review provides an updated overview of the latest evidence and societal guidelines on Lung Ultrasound (LUS), particularly in relation to medical Interstitial Lung Disease (ILD)”. So it is confusing because initially the main objective will be to support the application of LUS in clinical practice, but later on it seems that the manuscript will only review its application to interstitial disease, and not to ongoing integration in all aspects in routine clinical practice of LUS. Finally the revision includes mainly LUS findings in interstitial pathology so this seems to be the objective. But at the end, they speak about consolidations….  It would be advisable to carry out a thorough revision to clearly delimit the objectives of the study and to adjust the content of the manuscript to them.

Structure and development:  The document in its current form resembles more a practical manual on the use of ultrasound than an in-depth discussion on its implementation in clinical practice. So it makes some comments about its use in interstitial pathology (autoimmune disease, SARS-COV19, o SDRA…) and some references to its role in consolidations. If this is the content proposed, then perhaps a different structuration of the text would benefit the total content, allowing the reader to follow a line of progression, beginning with an explanation about basic lesions (B-lines, pleural line abnormalities, subpleural consolidations…) and progressing to its interpretation in different diseases and a description of the corresponding pattern in each condition. Some of the concepts are redundant, and appear in different places.  In my opinion the paper will improve in clarity, consistency and in quality (and finally would be more useful) with a restructuration of the content. It would add interest by focusing on implementation aspects (an organized definition of LUS findings in different pathologies), exploring potential challenges (some of them mentioned in introduction as absence of a validated format for reporting findings or need of a training program), benefits, and practical solutions.

Future perspectives: this section includes an important and controversial aspect referred to LUS, which is the best training method and how to evaluate the acquired skills. But this is the first reference to this issue in the whole manuscript, and it is a topic not related to the discussed content of the manuscript. The authors perhaps may consider discussing about it in the main text as part of the difficulties for the implementation of the tool.

Conclusions: The conclusions do not appear to be based on what has been presented throughout the text. For example, AI is not mention at all, but they conclude that it can be help “to solidify LUS as a cornerstone of pulmonary diagnostics in the future”. I suggest that the conclusions be revised to align closely with the topics developed in the paper.

Other comments:

I do not understand the utility of Figure 3.

All the LUS images include has “abdomen” as preset

A better reference to scores when they are mentioned, with a more specific and structured explanation of their application and content (Kigali score, LUS-ARDS score..) and perhaps some comments about the difficulties derived from the great variety of designed scores, the type of approximation of theses scores (quantitative or qualitative), it purpose in diagnosis or management or monitoring…

Author Response

REVIEWER #1:

Reply to comments:

Dear reviewer, I want to thank you for taking the time to read our work and make valuable comments. We are sure that they will greatly improve the text, making it of better quality for the journal and the reader.

C: Title: The current title does not adequately reflect the content nor does it provide an accurate idea of what the reader can expect to find in the text. I suggest rephrasing the title so that it better captures the focus and main objectives of the paper.

R: The title has been changed from "Lung Ultrasound Post-COVID-19: 'I'm a Fan, But It's Complicated!' Redefining the Discussion" to "Lung Ultrasound Post-COVID-19: A Pivotal Moment for Clinical Integration - Navigating Challenges and Seizing Opportunities." We believe this new title more accurately captures the essence of the work and the discussions that have been developed.

C: Objectives: The objectives are not sufficiently defined and, moreover, do not strictly correspond to the development of the manuscript. In the abstract,  the author presents this as objective:  “to advocate for the ongoing integration of LUS in routine clinical practice, particularly for diagnosing and monitoring lung diseases, and to emphasize its significant impact on patient outcomes following the COVID-19 pandemic”. Later on in methods it stated that “This narrative review provides an updated overview of the latest evidence and societal guidelines on Lung Ultrasound (LUS), particularly in relation to medical Interstitial Lung Disease (ILD)”. So it is confusing because initially the main objective will be to support the application of LUS in clinical practice, but later on it seems that the manuscript will only review its application to interstitial disease, and not to ongoing integration in all aspects in routine clinical practice of LUS. Finally the revision includes mainly LUS findings in interstitial pathology so this seems to be the objective. But at the end, they speak about consolidations….  It would be advisable to carry out a thorough revision to clearly delimit the objectives of the study and to adjust the content of the manuscript to them.

R: The objectives have been clarified and standardized throughout the manuscript, and the materials and methods section has been made clearer as well.

C: Structure and development:  The document in its current form resembles more a practical manual on the use of ultrasound than an in-depth discussion on its implementation in clinical practice. So it makes some comments about its use in interstitial pathology (autoimmune disease, SARS-COV19, o SDRA…) and some references to its role in consolidations. If this is the content proposed, then perhaps a different structuration of the text would benefit the total content, allowing the reader to follow a line of progression, beginning with an explanation about basic lesions (B-lines, pleural line abnormalities, subpleural consolidations…) and progressing to its interpretation in different diseases and a description of the corresponding pattern in each condition. Some of the concepts are redundant, and appear in different places.  In my opinion the paper will improve in clarity, consistency and in quality (and finally would be more useful) with a restructuration of the content. It would add interest by focusing on implementation aspects (an organized definition of LUS findings in different pathologies), exploring potential challenges (some of them mentioned in introduction as absence of a validated format for reporting findings or need of a training program), benefits, and practical solutions.

R: The introduction section has been optimized, clarified and reorganized according to the objectives.

A basic section has been added to highlight the difference between healthy and diseased subjects. The text has been structured and presented following the progression of the loss of aeration. The section that discusses the problems in the implementation and usefulness of artificial intelligence has been improved. As well as undergraduate and graduate training.

Finally, the text has been improved in clarity, quality consistency, and organization.

C: Future perspectives: this section includes an important and controversial aspect referred to LUS, which is the best training method and how to evaluate the acquired skills. But this is the first reference to this issue in the whole manuscript, and it is a topic not related to the discussed content of the manuscript. The authors perhaps may consider discussing about it in the main text as part of the difficulties for the implementation of the tool.

R: The section has been improved and placed under “controversies'”.

C: Conclusions: The conclusions do not appear to be based on what has been presented throughout the text. For example, AI is not mention at all, but they conclude that it can be help “to solidify LUS as a cornerstone of pulmonary diagnostics in the future”. I suggest that the conclusions be revised to align closely with the topics developed in the paper.

R: The conclusion has been improved to be aligned with the text and the changes made.

Other comments:

C: I do not understand the utility of Figure 3.

R: We agree that Figure 3 did not clearly contribute enough. Finally the images have been improved and image 3 has been removed.

All the LUS images include has “abdomen” as preset

R: All images were corrected

A better reference to scores when they are mentioned, with a more specific and structured explanation of their application and content (Kigali score, LUS-ARDS score..) and perhaps some comments about the difficulties derived from the great variety of designed scores, the type of approximation of theses scores (quantitative or qualitative), it purpose in diagnosis or management or monitoring…

R: We have expanded on the concepts behind the LUS score ARDS and the one used in Kigalis definition in order to make it more comprehensive for the reader.

Reviewer 2 Report

Comments and Suggestions for Authors

This is a review of the literature about the use of lung ultrasound (LUS) in diagnosing various diseases. The article focuses on interstitial lung diseases, COVID-19 pneumonia, ARDS and consolitations.

The title of the manuscript is misleading, by reading the title the reader expects a review that focuses only in the use of LUS in COVID-19 pneumonia. Please consider rephrasing.

Ιnclude in the text the limitations of the LUS in diagnosing interstitial lung dieases (eg the diagnostic accuracy is low, it only identifies disease that is sub-pleural, it cannot "see" lesions that are central) and the limitation of LUS in general (eg the diagnostic accuracy is reduced if the patient is obese). In the "Consolidation" section the use of LUS in distinguishing malignant from benign lesions is mentioned in one sentence, please be more elaborate. Please provide a table with the major findings of LUS in different lung diseases.

Author Response

REVIEWER #2:

Dear reviewer, I want to thank you for taking the time to read our work and make valuable comments. We are sure that they will greatly improve the text, making it of better quality for the journal and the reader.

C: This is a review of the literature about the use of lung ultrasound (LUS) in diagnosing various diseases. The article focuses on interstitial lung diseases, COVID-19 pneumonia, ARDS and consolitations.

C: The title of the manuscript is misleading, by reading the title the reader expects a review that focuses only in the use of LUS in COVID-19 pneumonia. Please consider rephrasing.

R: We agree that the title can be improved to better reflect the essence of the manuscript. The title has been changed from ''Lung Ultrasound Post-COVID-19: “I'm a Fan, But It's Complicated!” Redefining the Discussion'' to ''Lung Ultrasound Post-COVID-19: A Pivotal Moment for Clinical Integration - Navigating Challenges and Seizing Opportunities''.

C: Ιnclude in the text the limitations of the LUS in diagnosing interstitial lung dieases (eg the diagnostic accuracy is low, it only identifies disease that is sub-pleural, it cannot "see" lesions that are central) and the limitation of LUS in general (eg the diagnostic accuracy is reduced if the patient is obese). In the "Consolidation" section the use of LUS in distinguishing malignant from benign lesions is mentioned in one sentence, please be more elaborate. Please provide a table with the major findings of LUS in different lung diseases.

R: Line 100-116: The main general limitations of the LUS have been added.

R: The text about consolidations was reworded to be more comprehensive for the reader as you have suggested.

R: A table was added.

Reviewer 3 Report

Comments and Suggestions for Authors

To the Authors,

I want to thank the authors for their tremendous effort in preparing this research article, which fills the gap of an important research question and provides a detailed exploration of the Post-COVID-19 role of Chest Sonography and its applications,

The authors must address and respond to the following inquiries and requests.

  • Title:

Although it is attractive and informal—my opinion in this comment is advisory—I would prefer a more formal title such as The Role of Lung Ultrasound Post-COVID-19: Applications and Future Challenges or any equivalent.

  • In the abstract & introduction sections:
  • The abstract lacks clear 0bjective, methodology and summarized results. It could be improved
  • The introduction should thoroughly discuss the research question and hypothesis.
  • The gap in the literature on Lung Ultrasound post-COVID-19 should be highlighted early in the introduction.
  • Methodology:
  • The search strategy should be mentioned and discussed in detail.
  • You didn’t justify and discuss that only three researchers elicited the search could create selection bias.
  • Discuss the language restriction bias.
  • Although it’s a narrative review, you should state article inclusion and exclusion criteria.
  • Did you follow PRISMA guidelines?
  • Results:
  • The discussion and results sections are overlapping. Try to re-organize them.
  • Summarize the key trends in a paragraph or table
  • You rely on your results on literature rather than your vision and try to improve it,
  • Discussion section:
  • Highlight the role of LUS versus other modalities such as MSCT and MRI.
  • Discuss the cost-effectiveness and barriers to using LUS.
  • Postulate your vision in the future application of LUS.
  • Conclusion section:
  • Summarize your key findings
  • State the need and direction of future studies.
  • Limitations:
  • You should make a limitation section and address any Biase limitation that could have occurred and the study's narrative nature.
  • Acknowledge the need for further studies.

The article in its current form is not suitable for publication, and Major revisions are required. However, I highly recommend that the author address all the notes and complete the necessary data. A revised version may be accepted.

Author Response

REVIEWER #3

Dear reviewer, I want to thank you for taking the time to read our work and make valuable comments.

C: To the Authors,

I want to thank the authors for their tremendous effort in preparing this research article, which fills the gap of an important research question and provides a detailed exploration of the Post-COVID-19 role of Chest Sonography and its applications,

The authors must address and respond to the following inquiries and requests.

C: Title:

Although it is attractive and informal—my opinion in this comment is advisory—I would prefer a more formal title such as The Role of Lung Ultrasound Post-COVID-19: Applications and Future Challenges or any equivalent.

R: The title has been changed from ''Lung Ultrasound Post-COVID-19: “I'm a Fan, But It's Complicated!” Redefining the Discussion'' to ''Lung Ultrasound Post-COVID-19: A Pivotal Moment for Clinical Integration - Navigating Challenges and Seizing Opportunities''. We believe that this title better reflects the essence of the work and the discussion developed.

C: In the abstract & introduction sections:

The abstract lacks clear 0bjective, methodology and summarized results. It could be improved

The introduction should thoroughly discuss the research question and hypothesis.

The gap in the literature on Lung Ultrasound post-COVID-19 should be highlighted early in the introduction.

R: The abstract, methodology, discussion and conclusions have been improved and the information reorganized, as suggested in order to be more comprehensive for the reader.

C: Methodology:

The search strategy should be mentioned and discussed in detail.

You didn’t justify and discuss that only three researchers elicited the search could create selection bias.

Discuss the language restriction bias.

R: It is true that language can be a limitation, this has been clarified in the materials and methods and then discussed in the discussion as suggested.

C: Although it’s a narrative review, you should state article inclusion and exclusion criteria.

Did you follow PRISMA guidelines?

R: The inclusion and exclusion criteria have been developed in the materials and methods section and then discussed in the strengths and weaknesses of the study. We believe this will enhance this narrative review.

C: Results:

The discussion and results sections are overlapping. Try to re-organize them.

Summarize the key trends in a paragraph or table

You rely on your results on literature rather than your vision and try to improve it,

R: The manuscript has been reorganized and reformulated to make it clearer and more comprehensive for the reader.

C: Discussion section:

Highlight the role of LUS versus other modalities such as MSCT and MRI.

Discuss the cost-effectiveness and barriers to using LUS.

Postulate your vision in the future application of LUS.

R: The discussion has been reorganized and improved to include our vision of the LUS in the future. While we believe it is interesting to bring to the table the comparison of other tools such as MSCT, MRI or electrical impedance tomography, this would be beyond the scope and objectives of this review. However, we will consider this specific topic in the future.

C: Conclusion section:

Summarize your key findings

State the need and direction of future studies.

R: The final results have been summarized, and were arranged in key messages. The need for future studies has been suggested.

C: Limitations:

You should make a limitation section and address any Biase limitation that could have occurred and the study's narrative nature.

Acknowledge the need for further studies.

The article in its current form is not suitable for publication, and Major revisions are required. However, I highly recommend that the author address all the notes and complete the necessary data. A revised version may be accepted.

R: The limitations and strengths of the study have been spatially highlighted and discussed. Also the need to further studies.

Reviewer 4 Report

Comments and Suggestions for Authors

This study addresses the role of Lung Ultrasound (LUS) in clinical practice during and after the COVID-19 pandemic. Lung ultrasound (LUS) has been examined in detail for its importance in clinical decision-making processes over the last three decades. In addition, the role of LUS in the diagnosis of acute respiratory distress syndrome (ARDS) and interstitial lung diseases (ILD) is emphasized.

My suggestions for this article:

1) Reconsider the article title and abstract. Some statements are unclear. In addition, the abstract section is not fully comprehensive and is longer than necessary.

2) The writing style of the article should be improved. It should be written in a more academic style and some details should be expressed more clearly.

3) Standard reporting formats can be developed. It would be beneficial to suggest universal protocols for LUS reporting in the article.

4) More clinical application examples and case studies should be added. This contributes to making theoretical knowledge concrete.

5) The numerical breakdown of literature and research data should be increased. Adding more charts or data tables will increase the visibility of the analysis.

Comments on the Quality of English Language

-

Author Response

REVIEWER 4#

Dear reviewer, I want to thank you for taking the time to read our work and make valuable comments.

C:This study addresses the role of Lung Ultrasound (LUS) in clinical practice during and after the COVID-19 pandemic. Lung ultrasound (LUS) has been examined in detail for its importance in clinical decision-making processes over the last three decades. In addition, the role of LUS in the diagnosis of acute respiratory distress syndrome (ARDS) and interstitial lung diseases (ILD) is emphasized.

My suggestions for this article:

C: 1) Reconsider the article title and abstract. Some statements are unclear. In addition, the abstract section is not fully comprehensive and is longer than necessary.

R: The abstract has been modified and improved according to the reorganization of the main text.

C:2) The writing style of the article should be improved. It should be written in a more academic style and some details should be expressed more clearly.

R: The writing style has been improved and some sentences have been added to further develop certain aspects.

C:3) Standard reporting formats can be developed. It would be beneficial to suggest universal protocols for LUS reporting in the article.

R: A table has been added at the suggestion of the other reviewers including the main findings of the LUS in the objective scenarios. While two tables may be too many, it has been suggested that there is a need to develop and unify reporting formats.

C:4) More clinical application examples and case studies should be added. This contributes to making theoretical knowledge concrete.

R: At the suggestion of the reviewers, the manuscript has been clearly defined in the scenarios of Pneumonia, ILD and ARDS. We believe that while the reviewer is correct that adding more scenarios would make it a more complete review, we would be overstepping the objectives of the review and the search criteria used. However, we will consider discussing this specific topic in the future.

C:5) The numerical breakdown of literature and research data should be increased. Adding more charts or data tables will increase the visibility of the analysis.

R: A table has been added.

We hope that we have sufficiently addressed all of the reviewers’ comments, and will be happy to make further changes if required.

On behalf of all the authors, we extend our gratitude for your time spent in reading and analyzing our work.

Yours sincerely,

Round 2

Reviewer 3 Report

Comments and Suggestions for Authors

Thanks for submitting a well organised revised version of your work.